# ParaSMoE : Enabling Parallelism Hot-Switch for Large Mixture-of-Experts Models

## Abstract

Mixture-of-Experts (MoE) models has been demonstrated to be an effective paradigm for scaling Large language Model (LLM) parameters to hundreds of billions. A key consideration of MoE inference is parallelism strategy, which defines how parameters are distributed across multiple GPUs, and consequently dictates the communication pattern across the GPUs during model inference. We make an key observation that the optimal parallelism configuration is highly dependent on workload characteristics, which are dynamic in practice, shaped by different latency requirements in serving and by the decreasing number of active sequences in rollout phase of reinforcement learning (RL) . We introduce ParaSMoE that adapts the parallelism strategy to workloads. The core is an efficient "hot-switch" mechanism that seamlessly transitions between Expert Parallelism (EP) and Tensor Parallelism (TP), unleashing its ability to dynamically select the optimal parallelism for any given workloads. Through elaborated multi-level communication overlapping, Our experiments shows ParaSMoE can convert Qwen3-235B MoE model from EP to multiple TP instances in 0.7 seconds, with negligible memory overhead. We further project its potential to speedup batch generation in RL rollout phase by 1.4–3.7×.

## 1 Introduction

Mixture-of-Experts (MoE) (Shazeer et al., 2017) have emerged as a promising architecture in Large Language Models (LLM), achieving state-of-the-art performance by scaling model size to hundreds of billions parameters, without a proportional increase in computational cost. This is accomplished by sparsely activating a subset of "expert" sub-networks for each input token. The success of MoE is evident in its adoption by numerous state-of-the-art models, including Qwen3 (Yang et al., 2025), DeepSeek-V3 (DeepSeek-AI et al., 2025), and Grok-4 (xAI, 2025).

Efficient inference of MoE models requires careful deign, with model parallelism being a primary consideration. When the model size exceeds the memory capacity of a single GPU, it is necessary to distribute the model weights across multiple devices. For MoE inference, the two most prominent strategies are Tensor Parallelism (TP) which partitions the weights of individual weight matrices (Shoeybi et al., 2020a), and Expert Parallelism (EP) which distributes distinct experts across different GPUs (Rajbhandari et al., 2022; DeepSeek-AI et al., 2025). These two parallelism configurations present a fundamental trade-off. EP naturally scales to multiple GPUs, enabling it to sustain much higher batch size and provide better throughput under heavy request concurrency. However, this design incurs cross-server communication overhead, leading to higher per-token latency at small batch sizes. In contrast, TP excels in small-batch scenarios where it offers both lower latency and higher throughput within a single server. But its scalability is fundamentally constrained by limited device memory and significant communication costs under high request concurrency, making TP less efficient than EP. Therefore, the optimal parallelism is not static, but depends directly on workload characteristics.

Our systematic profiling of an industrial MoE inference system reveals two critical applications that inherently involve dynamic workload characteristics. (1) The first application is MoE serving with different requirements on per-token latency. Some requests with strict low-latency requirement cannot be satisfied by an EP serving instance (for example, 10 8xH100 machines for Deepseek V3), while deploying too many TP units can be wasteful under high request concurrency in general situations. Facing an influx of high priority requests with low latency requirement, a potential solution is to dynamically convert an inference unit from EP to TP, which offers lower latency compared to

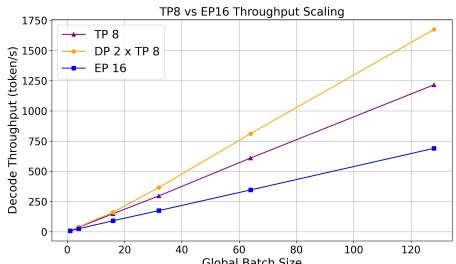 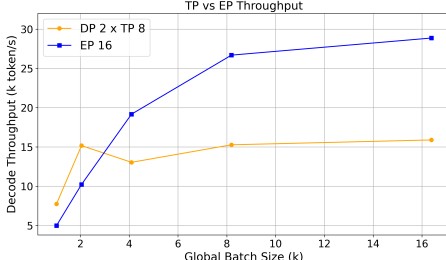

Figure 1: Comparison of Tensor Parallelism (TP) and Expert Parallelism (EP) on DeepSeek V2.5 236B with 2×8 H200 GPUs. The two plots illustrate their performance trade-off with small- and large-batch workloads. At a smaller batch size, TP yields better throughput than EP. However, EP scales linearly with concurrency and surpasses TP, which saturates early due to limited device count.

EP (although at a higher per-request serving cost). (2) The second application is batch generation for rollout in reinforcement learning (e.g. PPO (Schulman et al., 2017), GRPO (Shao et al., 2024), DAPO (Yu et al., 2025)). In every rollout step, batched inference begins with a large number of sequences. As generation progresses, sequences complete at different lengths, causing the effective batch size to gradually shrink (Hu et al., 2024; Sheng et al., 2025) as illustrated in figure 5 of Appendix C. Ideally, a batch generation could start in the high-throughput EP configuration, and dynamically switch to the configuration of multiple lower-latency TP instances with better aggregated throughput as the number of active sequences falls below a threshold. It helps to optimize resource usage for the remaining rollout process.

We introduce ParaSMoE to enable adaptive parallelism for MoE inference, significantly improving the efficiency of MoE inference system under varying workloads characteristics. The key innovation is a "hot-switch" mechanism (Ge et al., 2024) that transitions servers between EP and TP at runtime. This allows the inference framework to adapt its parallelism configuration on-the-fly, selecting the most efficient strategy for the current workload without the need to restart or reconfigure the inference system. The parallelism conversion is highly efficient and can complete under 0.7s on Qwen-3 235B model. We designed a sophisticated multi-level communication overlapping scheme, which carefully orchestrates intra-node (e.g., NVLink) and inter-node (e.g., InfiniBand and Amazon EFA (AWS, 2025)) communication to minimize system stalls during the conversion.

To the best of our knowledge, ParaSMoE is the first work that enables practical dynamic parallelism in MoE inference systems (Kwon et al., 2023; Zheng et al., 2024). We provide a comprehensive analysis of the switch and an efficient communication overlap strategy in Section 3.3. In Section 4.2, we experiment with our method using Qwen-3-235B on two 8xH200 servers, demonstrating that ParaSMoE 's efficacy of switching parallelism configurations at a minimal time cost. Then we analyze its ability to enhance MoE serving with quality of service requirement and potential to improve batch inference by 1.4 - 3.7× in rollout phase of reinforcement learning in Section 4.3.

## 2 Parallelism in MoE Inference

We first briefly describe TP and EP, and compare their pros and cons with regard to the following key factors that dictate the performance of a MoE inference system.

- **Weight Reading Amortization.** The model weights have to be loaded from GPU HBM to on-chip SRAM for one decoding iteration. Such expensive model weight loading cost is amortized over the entire batch, thus a large batch size could effectively amortize the loading cost and reduce the overall serving cost.
- **Communication Cost.** Intra-node communication is through high-speed networsk like NVIDIA NVLink, but cannot expand to more than 8 GPUs. Inter-node communication (via Infiniband or Amazon EFA) is significantly slower than intra-node, but can expand to a large number of nodes and GPUs.
- **Device Memory Capacity.** Each request generates key-value (KV) cache that resides in GPU memory. Once device memory is saturated by existing caches, no additional requests can be admitted. Thus, the maximum batch size that the system can support is ultimately bounded by the aggregate memory capacity of the devices.

**Tensor Parallelism (TP)** partitions the weight matrices of each model layer across devices, typically within a single multi-GPU server. Recent advancements, such as the large HBM volumes on latest GPUs and quantization techniques like FP8/FP4 (Kuzmin et al., 2024; DeepSeek-AI, 2025a; Wang et al., 2025), have made it feasible to fit even large MoE models onto a single server with 8 GPUs. In this setup, the communication cost of TP, which involves `All-Reduce` operations within each layer, is minimized by high-speed intra-node connectors like NVLink. Coupled with the fact that workload is evenly shared across multiple GPUs, TP is highly suitable for latency sensitive small-batch inference. However, the throughput is bounded by the expensive `All-Reduce` as batch size grows large. An important advantage of TP compared to EP is in its uniform distribution of compute, where each GPUs is doing exact the same amount of computing. EP suffers from the unbalanced expert issue, where tokens are concentrated to a few experts and the other experts are sitting idle. On the downside, extending TP across multiple nodes is generally considered impractical due to the prohibitive cost of `All-Reduce` operations over the inter-node network. This confines TP to single-node inference, imposing a ceiling on the batch size with limited device memory, as there is limited memory space to store the KV cache and intermediate activation values for a larger batch size. Consequently, the cost of loading the model parameter from HBM to SRAM can only be amortized over a constrained batch size, fundamentally restricting TP's throughput and per-token serving cost.

**Expert Parallelism (EP)** partitions the model by distributing experts (FFN modules) across different GPUs. Once the routing layer selects an expert for a given token, the token is routed to the device hosting the expert. The tokens routed to a particular expert are batched and processed by the expert (`dispatch`), then routed back to each token's originating GPU (`combine`) via highly optimized `All-to-All` communication (Zhao et al., 2025a; Perplexity-AI, 2025). While the attention module is replicated following a data parallel pattern (with optional ), EP unlocks the model's ability to deploy at scale. An inference unit can span tens of 8 GPU nodes, leaving plenty of memory space to hold KV cache of a large batch and amortize the weights loading and other system overheads. This is in clear contrast to the TP case where an inference unit can not expand over multiple-nodes. This is among the key reasons for the success of DeepSeek series models. For small batches, however, the inter-node communication latency bottlenecks the processing time, making EP an inferior choice and leading to under-utilization of GPUs.

Our experiments further illustrate the complementary strengths of EP and TP with varying workload regions. As shown in Figure 1, under small-batch, low-concurrency scenarios, two TP instances of 8 GPUs within a single node achieve higher throughput as well as lower latency compared to 16 GPUs using EP. This is due to TP's low-latency intra-node communication and uniform compute distribution. EP suffers in this region, as the overhead of inter-node `All-to-All` communication dominates when amortized over a small batch. In contrast, EP scales significantly better under high-concurrency workloads, where large batches amortize EP's communication overhead and enable superior throughput. While TP's throughput saturates early due to its limited device count, EP's throughput continues to scale linearly with the number of requests, eventually outperforming TP by a large margin. These experiments confirm that a static parallelism choice inevitably under-utilizes resources across fluctuating workloads, and motivate the need for an adaptive system like ParaSMoE to dynamically switch between EP and TP.

## 3 Weights Resharding

While model forward process remains almost the same, the primary challenge to achieving fast parallelism switch lies in the data plane, where the massive weights of the MoE layers must be re-sharded and re-distributed across devices. ParaSMoE supports bi-directional re-sharding between EP and TP. Since the two directions are conceptually symmetric, we focus on the more complex EP-to-TP case as a representative example.

### 3.1 Cluster and Model

A GPU cluster today generally consists of $N$ servers, each equipped with $P$ devices, for a total of $G = N \times P$ devices. Within every server, devices are connected by NVLink. Devices communicate across servers via remote direct memory access (RDMA) network. In practice, EP usually takes the whole cluster but TP is constrained to a single server. Therefore, we assume a EP degree of $G$ and a TP degree of $P$ during the switch. All devices in the cluster initially compose a single EP instance. After the switch, the devices in each server operate synchronously under TP while different servers work under the data parallelism (DP) pattern. Figure 2 illustrates a cluster of 2 servers ($N = 2$)

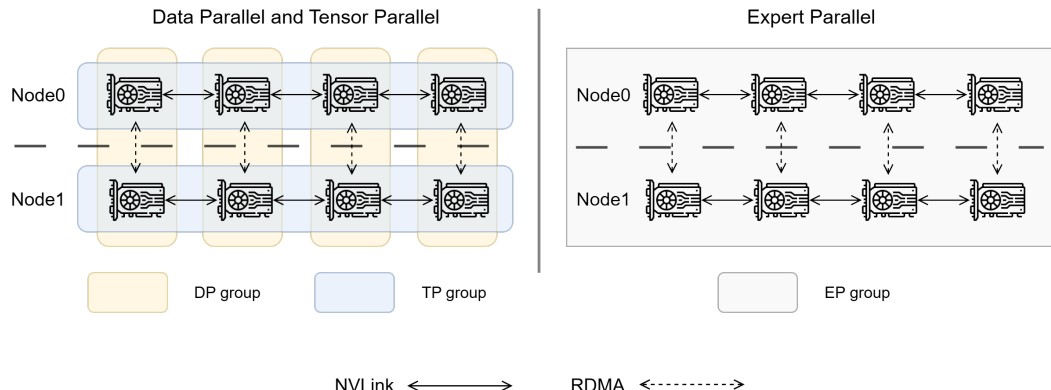

Figure 2: Cluster topology and different parallelism configurations. Communication groups are pre-initialized to enable hot-switching. **Left**: intra-node Tensor Parallelism (TP) with inter-node Data Parallelism (DP). **Right**: global Expert Parallelism (EP) across all devices.

equipped with 4 devices each ($P = 4$). These two servers can either work together under EP or separately under TP.

Corresponding to figure 2, we denote three types of communication groups at different scopes:

- $\epsilon$: the global expert parallel (EP) group, spanning all GPUs in the serving cluster. EP collectives use `AlltoAll` over $\epsilon$ to route tokens to experts across the entire cluster.
- $\tau$: intra-node tensor parallel (TP) groups. Each group is typically confined to a single node and executes `AllReduce` to synchronize partial results.
- $\delta$: cross-node data parallel (DP) groups. These connect GPUs of the same rank across all TP instances and are used for communication over the inter-node network.

Without loss of generality, a MoE model can be specified by the number of experts $E$, the hidden dimension $H$, and the intermediate dimension $I$. Under expert parallelism (EP), each GPU hosts $L$ local experts such that $E = G \times L$, where $G$ is the total number of GPUs and $E$ is assumed divisible by $G$. Based on this formulation, we next describe how to transform layer weights during the transition from EP to TP.

### 3.2 Weights Transformation

In existing serving systems, weight matrices are typically stored in column-major order to match the requirements of high-performance BLAS libraries (Goto & Geijn, 2008). This convention places the output dimension before the input dimension in the memory layout. To align with this practice, we follow the semantics that each weight matrix is stored with a column-major order when describing the transformation. Meanwhile, weights fusion (Aminabadi et al., 2022; Kim et al., 2023) is widely adopted to merge serval weight matrices into one and reduce kernel launch overheads. For simplicity, we do not expand the fused dimensions as they only introduce another dimension for permutation without adding additional operations.

We begin by defining the objective of the transformation. After switching from EP to TP, each TP group must hold a complete copy of the model weights and thus be capable of functioning as an independent serving instance. Concretely, a GPU with EP rank $r_\epsilon$ transitions to the corresponding TP rank $r_\tau$ within its TP group.

Modern MoE architectures follow a unified layer structure, in which each layer consists of an attention module and an MoE module. To describe the transformation procedure, we focus on a single layer, noting that the same process applies uniformly across the entire model. The weights within a layer can be divided into two categories: attention weights and expert weights. Each category requires a distinct transformation strategy, which we detail in the following subsections. We first discuss the comparatively simple case of attention weights, before turning to the more complex transformation of expert weights.

**Attention weights.** In an attention module, the transformation of projection weights from EP to TP is comparatively straightforward. Let $W^{qkv}$ and $W^o$ denote the QKV projection and output

weight matrices, respectively. Under EP, these weights are fully replicated across all devices, so the transition to TP requires only selecting the slice corresponding to the target TP rank $r_\tau$ and discarding the remainder.

In TP, the QKV projection matrix $W^{qkv}$ is column-partitioned. To align with this layout, we first transpose $W^{qkv}$ so that the partitions assigned to different TP ranks are stored contiguously, after which the appropriate slice $W^{qkv}_{r_\tau}$ is retained. This permutation step is necessary to ensure contiguous memory access and thereby achieve optimal GEMM performance. By contrast, the output projection $W^o$ naturally conforms to TP's row-partitioned layout. Each slice is already stored contiguously in memory, so no permutation is required during the transformation.

**Expert weights.** The transformation of expert weights is more intricate than that of attention weights. Let the weights of a MoE module be denoted by $W^e$, which contains an up-projection matrix $U$ and a down-projection matrix $D$ for each expert. Suppose the model has $E$ experts. We denote the up- and down-projection matrices of each expert as $U_0, U_1, \ldots, U_{E-1}$ and $D_0, D_1, \ldots, D_{E-1}$, respectively.

With EP, experts are evenly distributed across $G$ devices, so that each device stores $L$ consecutive experts. For instance, the device with EP rank $r_\epsilon$ holds experts $W^e_{r_\epsilon L}, \ldots, W^e_{(r_\epsilon+1)L-1}$. On device $r_\epsilon$, the up-projection weights can be expressed as a tensor $U_{r_\epsilon}$ of shape $[L, I, H]$, where $I$ and $H$ denote the input and hidden dimensions. After transforming to tensor parallelism (TP), the device with TP rank $r_\tau$ holds a slice of all experts, represented by another tensor $U_{r_\tau}$ of shape $[E, I', H]$, where $I' = \frac{I}{P}$ and $P$ is the TP degree. To make the relationship between the two layouts explicit, we expand both tensors as:

$$U_{r_\epsilon} : [L, P, I', H] \quad U_{r_\tau} : [G, L, I', H]$$

Recalling that $G = N \times P$ (where $N$ is the number of data-parallel groups), we can further expand the TP representation as:

$$U_{r_\tau} : [N, P, L, I', H]$$

Intuitively, under EP each device stores only a small subset of experts, whereas under TP each device must hold a slice of all experts. This requires gathering and redistributing weights across devices. The transformation proceeds in four steps:

*1. All-Gather across DP groups.* Since each $\tau$-ranked device needs slices from all experts, devices first perform an all-gather across data-parallel groups. This step increases the number of experts per device from $L$ to $L \times N$, so that every machine collectively holds all $E$ experts. After the operation, the weights are:

$$U_{r_\epsilon} \xrightarrow{\text{all-gather}} U'_{r_\epsilon}, \quad \text{where } U'_{r_\epsilon} : [N, L, P, I', H] \tag{1}$$

*2. Permutation for contiguity.* To prepare the tensor for efficient redistribution, we permute $U'_{r_\epsilon}$ so that slices of different experts become contiguous:

$$U'_{r_\epsilon} \xrightarrow{\text{permute}} U''_{r_\epsilon}, \quad \text{where } U''_{r_\epsilon} : [P, N, L, I', H] \tag{2}$$

*3. All-to-All across TP groups.* Next, an all-to-all is performed within each TP group to redistribute slices across devices. Although the tensor shape is preserved, the weights are reassigned so that each device $r_\tau$ now holds the correct TP slice of all experts:

$$U''_{r_\epsilon} \xrightarrow{\text{all-to-all}} U'_{r_\tau}, \quad \text{where } U'_{r_\tau} : [P, N, L, I', H] \tag{3}$$

*4. Final Permutation.* Finally, we permute the tensor again to restore the canonical order of experts:

$$U'_{r_\tau} \xrightarrow{\text{permute}} U_{r_\tau}, \quad \text{where } U_{r_\tau} : [N, P, L, I', H] \tag{4}$$

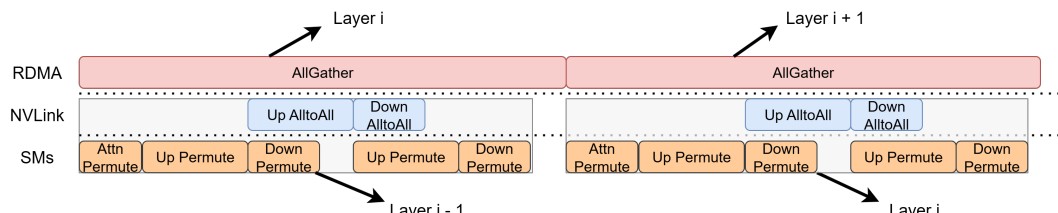

Figure 3: The execution pipeline for layer-wise weight re-sharding. The long latency of the inter-node `All-Gather` over RDMA is used to hide the execution of the `Permute` and the intra-node `All-to-All` over NVLink for the preceding layer. A MoE module contains $U$ and $D$ projection weights, where the size of $U$ weights is twice of $D$.

Through these four steps, each device transitions from storing a small consecutive block of experts (EP layout) to storing a partial slice of all experts (TP layout). Importantly, all devices perform the same sequence of operations in parallel under a single-program multiple-data (SPMD) execution model, ensuring symmetry and scalability.

The transformation of down-projection weights $D$ largely mirrors the procedure described for the up-projection $U$, involving the same sequence of all-gather, permutation, all-to-all, and final permutation operations. However, the key difference lies in the tensor layout: $D$ has an initial shape of $[L, H, I]$, and under TP it is row-partitioned along the output dimension $I$. As a result, all redistribution operations must slice along the $I$ dimension, rather than the $H$ dimension as in the case of $U$. Consequently, the permutation order differs to guarantee that each TP slice of $D$ remains contiguous in memory and compatible with GEMM execution. To avoid redundancy in the main text, we defer the detailed derivation and equations for $D$ to Appendix B.

### 3.3 Pipelined Execution

For the EP-to-TP transition, the re-sharding process for each MoE layer is decomposed into three hardware-specific operations which are detailed in Section 3.2:

- Inter-Node All-Gather: All expert weights are first collected onto each node using an `All-Gather` operation over DP groups. Since DP groups span across nodes, this collective leverages the inter-node RDMA network to exchange parameters, resulting in each node obtaining a full replica of all experts.
- Weight Permutation: Once the weights are locally available, they must be rearranged from the EP layout into the TP-compatible layout. This permutation, which reorganizes the memory layout of weight tensors, is executed by the GPU's streaming multiprocessors (SMs).
- Intra-Node All-to-All: After permutation, the weights are partitioned into TP slices and distributed to the target devices within each TP group using an `All-to-All` communication over NVLink. After this step, each GPU holds exactly its TP-responsible shard of all experts.

A key design choice in ParaSMoE is to avoid executing these steps sequentially. Instead, we pipeline them across layers to maximize utilization of the interconnects and compute resources. As shown in Figure 3, the high-latency inter-node `All-Gather` provides a natural window to overlap the lower-latency permutation and `All-to-All` of preceding layers. For instance, while the RDMA transfer for layer $i$ is still in progress, the GPU's SMs and NVLink fabric are simultaneously finalizing the re-sharding of layer $i - 1$. This staggered schedule ensures that communication and computation resources are continuously occupied without long idle phases.

Beyond MoE experts, attention modules also require transformation during EP-to-TP switching, but their structure simplifies the process. Each attention block contains $W^{qkv}$ and $W^o$, which are not sharded across experts but do require a layout change. For these weights, only one permutation is needed. We therefore schedule attention weight permutations at the start of each layer, preceding the permutation of $U$. This ensures that attention parameters are always ready in TP layout by the time token processing resumes, without introducing additional synchronization points.

Finally, we also pipeline within the MoE layers themselves. Each MoE block contains both $Up$ and $Down$ projection weights; the $Up$ projections are twice the size of $Down$. To better utilize NVLink bandwidth and SM compute cycles, we overlap the permutation and intra-node `All-to-All` for these two weight matrices. This fine-grained scheduling strategy not only minimizes end-to-end

transition latency but also achieves balanced utilization across RDMA, NVLink, and GPU compute, leading to the sub-second hot-switch performance demonstrated in Section 4.2.

**Theoretical cost.** We analyze the end-to-end switching cost under the communication-bound region where the *inter-node All-Gather* dominates and all other steps can be fully overlapped as illustrated in figure 3. Assume $N$ nodes, each with $P$ GPUs; the model has $M$ parameters stored in data type of size $\beta$ bytes. During the EP-to-TP transformation, each node needs to gather the remaining $(N-1)$ replicas of its shard. Since the effective shard per node is $M/(NP)$, the volume a node must receive is $\frac{M\beta}{NP}(N-1)$. Under the bandwidth-bound assumption (unit bandwidth normalized for clarity), the switching time is therefore

$$T_{\text{switch}} \; = \; \frac{M\beta}{NP}\,(N-1). \tag{5}$$

If we account for an explicit inter-node network bandwidth $B_{\text{inter}}$, this becomes

$$T_{\text{switch}} \; = \; \frac{M\beta}{NP\,B_{\text{inter}}}\,(N-1) \; = \; \frac{M\beta}{P\,B_{\text{inter}}}\Big(1-\frac{1}{N}\Big) \tag{6}$$

which shows that the cost has an upper bound of $\frac{M\beta}{P}$. Intuitively, adding nodes shrinks each node's shard size by a factor of $1/N$, while the `All-Gather` fan-in grows only linearly with $(N-1)$; these effects nearly cancel, yielding a cost that does *not* grow substantially with cluster size. Under our overlapped schedule, permutation and intra-node collectives remain hidden behind the inter-node All-Gather, so the parallelism transition time is **invariant** with cluster size. This is a surprising and high desirable result, ensuring our method can be scaled to inference unit with very large number of nodes and GPUs.

## 3.4 Memory Overhead

We analyze the memory overhead of expert weights during the EP-to-TP transition, as expert parameters dominate the overall model size. The analysis is conducted per device and applies symmetrically to all devices in the cluster. Consider a MoE model with $K$ layers, each containing $E$ experts. For each expert, the feed-forward block consists of an *Up* projection of size $2H \times I$ and a *Down* projection of size $H \times I$, yielding a total of $3HI$ parameters per expert. Let $\beta$ denote the number of bytes per parameter (e.g., $\beta=2$ for BFloat16).

**Steady-State Memory Overhead.** The per-layer memory footprint under Expert Parallelism (EP) and Tensor Parallelism (TP) is:

$$M_{\text{EP}}^{\text{layer}} \; = \; \beta \cdot \frac{E}{G} \cdot 3HI = \beta \cdot \frac{E}{NP} \cdot 3HI \qquad M_{\text{TP}}^{\text{layer}} \; = \; \beta \cdot \frac{E}{P} \cdot 3HI$$

Aggregated across $K$ layers, the memory consumption sums up to be:

$$M_{\text{EP}}^{\text{model}} \; = \; K\beta \cdot 3HI \cdot \frac{E}{G} \qquad M_{\text{TP}}^{\text{model}} \; = \; K\beta \cdot 3HI \cdot \frac{E}{P}$$

**Intermediate Buffers and Transition Overhead.** During the transition, additional buffers are required. Two alternating buffers suffice for the current layer $i$'s re-sharding, and one more is needed to pre-fetch layer $i+1$, each of size $\beta \cdot \frac{E}{P} \cdot 3HI$. At step $i$, layers $\{0,\ldots,i-1\}$ have already been transformed into TP, while layers $\{i+1,\ldots,K-1\}$ remain in EP. The instantaneous memory overhead is therefore:

$$M(i) \; = \; \beta \cdot 3HI \left[\frac{E}{P}\,(i+3) \; + \; \frac{E}{G}\,(K-i-1)\right] \; = \; \beta \cdot 3HI \cdot \frac{E}{P}\left[(i+3) \; + \; \frac{K-i-1}{N}\right] \tag{7}$$

As $i$ increases, $M(i)$ grows monotonically, reaching its maximum at $i=K-1$. The peak memory overhead during transformation is thus:

$$M_{\max} \; = \; \beta \cdot 3HI \cdot \frac{E}{P}\,(K+2)$$

Comparing with the steady-state TP memory overhead:

$$\frac{M_{\max}}{M_{\text{TP}}^{\text{model}}} = \frac{K+2}{K} = 1 + \frac{2}{K} \tag{8}$$

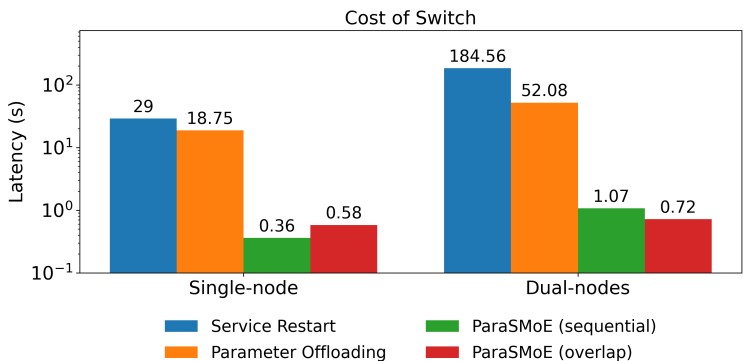

Figure 4: Latency of hot-switching across single-node (8×A100) and dual-node (two 8×H200) deployments. While service restart and parameter offloading incur tens of seconds of overhead, ParaSMoE reduces the cost to the sub-second regime, enabling practical adaptive parallelism.

showing that the maximum memory overhead is only marginally larger than the TP steady state, with an overhead that diminishes as number of layers increases. Large MoE models such as DeepSeek V3 (DeepSeek-AI et al., 2025), Qwen3 (Yang et al., 2025) and Kimi K2 (Team et al., 2025) typically have more than 50 layers, making the additional memory footprint **less than 5% of model memory**. If we include the KV cache and intermediate activation memory into consideration, the memory overhead is even more negligible.

# 4 Evaluation

## 4.1 Experimental Settings

We evaluate ParaSMoE on two hardware environments representative of single-node and dual-nodes inference deployments. All experiments use BFloat16 precision for model parameters.

**Single-node (8×A100).** Our first set of experiments is conducted on a single server equipped with eight NVIDIA A100 GPUs, each with 40 GB of memory, connected through NVLink. We deploy the Qwen3-MoE-30B model and configure it initially under an expert parallelism (EP) layout with 8-way partitioning (EP8). We then perform a hot-switch to a hybrid configuration combining tensor parallelism (TP4) with data parallelism (DP2). All collective communications in this setup are intra-node and utilize NVLink.

**Dual-nodes (2×8 H200).** We further evaluate ParaSMoE at larger scale on a distributed setting with two servers, each containing eight NVIDIA H200 GPUs. The servers are interconnected with a high-speed 3,200 Gbps RDMA-capable network, while intra-node communication is supported by NVLink. On this system, we deploy the significantly larger Qwen3-MoE-235B model. The baseline configuration uses EP16 across the 16 GPUs, and we perform a hot-switch to a hybrid DP2–TP8 layout. This transformation involves both intra-node and inter-node collectives, thus stressing system's ability to overlap communication and computation.

**Baselines.** We compare ParaSMoE against two baseline approaches commonly used in practice. The first baseline, service restart, switches parallelism by terminating the current process and reinitializing the system with a new parallelism configuration. To avoid a completely cold start, we ensure that the model weights remain hot-cached in the file system. This method is simple to implement but incurs long delays since all parameters must be reloaded from disk and communication groups reconstructed. The second baseline, parameter offloading, avoids a full restart by temporarily offloading model parameters to host memory, reconfiguring the communication groups, and then reloading parameters back to the GPUs. While faster than a full restart, this approach still requires moving massive amounts of data across the PCIe bus, which can introduce substantial latency.

## 4.2 Cost of Switch

Figure 4 presents the latency of hot-switching across both single-node and dual-node deployments. Restarting the service and parameter offloading incur prohibitive overheads that span tens of seconds. In contrast, ParaSMoE consistently drives the transition cost down to the **sub-second** regime, making dynamic adaptation feasible in practice.

On the single-node setup, the model is relatively small and thus cannot fully exploit the available NVLink bandwidth. Since all communication (All-Gather and All-to-All) is confined within the

NVLink fabric, the traffic competes for the same resources, limiting the benefit of overlapping communication and computation. Consequently, the overlapped pipeline offers worse latency than the sequential approach in this setting. By contrast, the dual-node experiment involves a much larger model and introduces inter-node communication across a high-bandwidth network. Here, the larger data volume fully utilizes the network bandwidth, and the fine-grained pipelined scheduling of ParaSMoE is able to hide much of the latency. According to the analytical model in section 3.3, the theoretic switch cost is around 0.6 seconds. ParaSMoE can achieve above 80% of the optimal performance, showcasing that overlap is highly effective, yielding significant additional speedup.

### 4.3 Benefit of ParaSMoE in practical MoE serving

**MoE Serving with Varying Traffic**    In real-world serving, requests often differ in both latency sensitivity and arrival rate. Enterprise customers may demand strict latency guarantees, while general user traffic benefits more from high throughput. Moreover, request rates fluctuate over time, making a fixed parallelism strategy ineffective at service level. ParaSMoE enables sub-second hot-switching between TP and EP, allowing the system to adapt its execution mode to current workload conditions. This flexibility ensures low latency for critical queries while sustaining high throughput during traffic surges, providing an effective mechanism to meet diverse quality-of-service (QoS) requirements.

**Batch generation in Reinforcement Learning**    Current reinforcement learning training frameworks employs a static parallelism configuration for the batch inference in the rollout phase. Every batch features both numerous decoding tokens and long tail problems, which makes traditional multiple TP deployment suffer in the beginning and EP suffer towards the end of rollout for large MoE models. However by bridging the gap between TP and EP, ParaSMoE is able to combine their advantages and consistently retain a relatively high throughput in any situation with a minimal switch cost. Through estimation detailed in appendix C, it is able to reduce the rollout time by 1.4 to 3.7x compared with TP or EP deployment alone while the final speedup eventually depends on workloads and cluster specifications.

## 5  Related Works

Research on Mixture-of-Experts (MoE) spans model design, parallelism strategies, and inference optimizations. The sparsely-gated MoE layer(Shazeer et al., 2017) enabled large-scale models such as DeepSeek-V3 (DeepSeek-AI et al., 2025) and Qwen3 (Yang et al., 2025) that scale efficiently, but their massive size complicates inference.

Parallelism methods are central to MoE deployment. Tensor parallelism (TP) (Shoeybi et al., 2020b) offers an optimal latency within multiple parallel devices, whereas expert parallelism (EP) (Rajbhandari et al., 2022) provides higher throughput at scale. Recent works (Liu et al., 2025; Singh et al., 2023) explore a hybrid parallelism configuration. ParaSMoE builds on these by enabling dynamic hot-switch between them.

Recent works address adaptivity and system efficiency. Some explore hot-switching during training (Ge et al., 2024), while Tutel (Hwang et al., 2023), HAP (Lin et al., 2025) investigate adaptive or hybrid strategies. Additional efforts target inference optimization through communication reduction (Zhao et al., 2025b; Perplexity-AI, 2025), load balancing (DeepSeek-AI, 2025b), expert buffering (Huang et al., 2024), and kernel specialization (DeepSeek-AI, 2025a). These approaches improve efficiency under fixed layouts, while ParaSMoE uniquely contributes sub-second, workload-aware switching between EP and TP for dynamic inference.

## 6  Conclusion

We presented ParaSMoE , a novel method that dynamically converts parallelism configurations for MoE inference. We identified a series of operations that efficiently achieves parallelism conversions, and carefully orchestrated the intra-node and inter-node communications to minimize the stall time. We demonstrated the parallelism flexibility enabled by our method could be greatly improve inference efficiency and hardware utilization in real world inference tasks, namely inference with Quality-of-Service requirement, and batched inference for reinforcement learning.

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

## A   The Use of Large Language Models (LLMs)

In preparing this submission, we made limited use of large language models (LLMs) as assistive tools. Specifically:

- **Writing and Editing:** LLMs were used to aid in polishing the writing, including rephrasing sentences for clarity, improving grammar, and refining overall readability. All technical content, research contributions, and core ideas originated from the authors.
- **Retrieval and Discovery:** LLMs were used to assist in retrieving relevant literature and discovering related work. The final selection, interpretation, and integration of related work were conducted solely by the authors.

LLMs did not contribute to research ideation, system design, experiments, or analysis. The authors take full responsibility for all content presented in this paper.

## B   Transformation of Down-Projection Weights

For completeness, we detail the EP-to-TP transformation for the down-projection weights $D$. The overall sequence of operations mirrors that of the up-projection $U$, but differs in the slicing dimension due to the memory layout.

Initially, on an EP rank $r_\epsilon$, the down-projection weights are stored as:

$$D_{r_\epsilon} : [L, H, I]$$

where $L = \frac{E}{G}$ is the number of experts per device, $H$ is the hidden dimension, and $I$ is the input dimension. After transforming to TP, the target tensor for TP rank $r_\tau$ should have the shape:

$$D_{r_\tau} : [E, H, I'], \quad \text{where } I' = \frac{I}{P}$$

We expand these tensors to make the correspondence explicit:

$$D_{r_\epsilon} : [L, H, P, I'], \quad D_{r_\tau} : [G, L, H, I']$$

Since $G = N \times P$, we further expand the TP layout as:

$$D_{r_\tau} : [N, P, L, H, I']$$

The transformation proceeds in four steps:

1. **All-Gather across DP groups** Devices gather all expert weights across data-parallel groups, increasing the number of experts per device from $L$ to $L \times N$:

$$D_{r_\epsilon} \xrightarrow{\text{all-gather}} D'_{r_\epsilon}, \quad D'_{r_\epsilon} : [N, L, H, P, I']. \tag{9}$$

2. **Permutation for contiguity** We permute the tensor so that slices along the $I$ dimension are contiguous, enabling efficient redistribution:

$$D'_{r_\epsilon} \xrightarrow{\text{permute}} D''_{r_\epsilon}, \quad D''_{r_\epsilon} : [P, N, L, H, I']. \tag{10}$$

3. **All-to-All across TP groups** An all-to-all communication redistributes slices across TP ranks. Although the shape remains unchanged, the contents are reassigned so that each TP rank $r_\tau$ obtains the correct slice of all experts:

$$D''_{r_\epsilon} \xrightarrow{\text{all-to-all}} D'_{r_\tau}, \quad D'_{r_\tau} : [P, N, L, H, I']. \tag{11}$$

4. **Final Permutation** Finally, the tensor is permuted to restore the canonical ordering of experts:

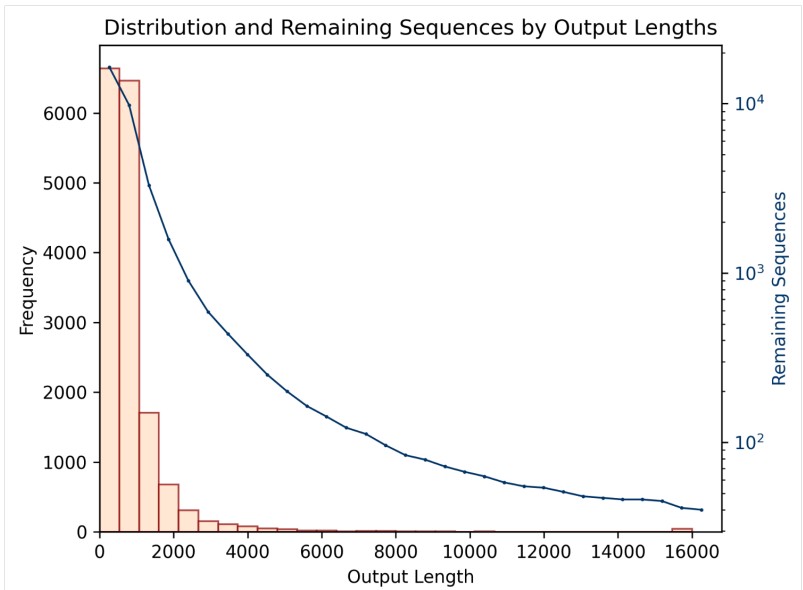

Figure 5: Distribution of output length in a batch. The curve presents the shrinking batch size over output length. The batch contains 16,382 decoding requests in total, where most sequences terminate within 2k tokens, while a small fraction extends up to 16k tokens, creating a long-tail workload imbalance.

$$D'_{r_\tau} \xrightarrow{\text{permute}} D_{r_\tau}, \quad D_{r_\tau} : [N, P, L, H, I']. \tag{12}$$

Through these steps, each device transitions from storing a small consecutive subset of experts (EP layout) to storing a partial slice of all experts along the $I$ dimension (TP layout).

## C    Estimated Improvement for Rollout in Reinforcement Learning

To approximate the speedup of ParaSMoE relative to static Expert Parallelism (EP) and Tensor Parallelism (TP), we construct a cost model based on reported throughput and latency characteristics.

**High-Concurrency Throughput.**    For large-scale Mixture-of-Experts (MoE) reinforcement learning, clusters with a few hundred devices enable high EP degrees (e.g., EP 64). Under such conditions, SGLang's report (SGLang, 2025) on DeepSeek V3 shows that EP achieves $5\times$ higher throughput (around 22k tokens/s) than TP (around 4.5k tokens/s). We denote the measured throughput of EP and TP under high concurrency as $T_{\text{EP}}$ and $T_{\text{TP}}$, respectively (in tokens/s).

**Low-Concurrency Latency.**    When the batch size shrinks after most requests have finished, the system transitions into a latency-dominated regime. Based on public reports, we approximate the per-token latency of TP as $\ell_{\text{TP}} = 20$ ms and that of EP as $\ell_{\text{EP}} = 50$ ms.

**Workload Characterization.**    A workload is described by:

- $N$: total number of requests in the batch,
- $L_{\text{maj}}$: major length where most requests terminate,
- $L_{\text{max}}$: maximum length among all requests.

Figure 5 illustrates an example from the DAPO dataset containing $N{=}16$k requests. The distribution shows that most of sequences finish before $L_{\text{maj}}{\approx}2$k tokens, while only a small fraction continues toward the long tail with $L_{\text{max}}{=}16$k. This heavy skew leads to a rapid decrease in active batch size as decoding progresses, posing challenges for efficient parallelism due to the mismatch between early-stage and tail-stage workloads.

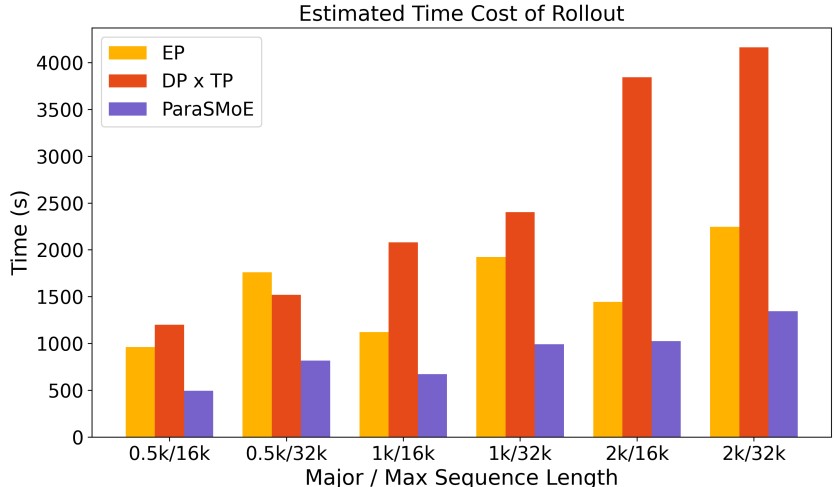

Figure 6: Estimated rollout time under different major and maximum lengths. ParaSMoE consistently outperforms static EP and DP×TP by adapting between throughput-dominated and latency-dominated phases.

**Cost Model.** The total serving cost is modeled as the sum of a throughput-bounded component (before $L_{\mathrm{maj}}$) and a latency-bounded component (after $L_{\mathrm{maj}}$):

$$C_{\mathrm{total}} \;=\; C_{\mathrm{throughput}} + C_{\mathrm{latency}}.$$

The throughput-bounded cost is

$$C_{\mathrm{throughput}} \;=\; \frac{N \cdot L_{\mathrm{maj}}}{T},$$

where $T \in \{T_{\mathrm{EP}}, T_{\mathrm{TP}}\}$ is the effective throughput under high concurrency.

The latency-bounded cost is

$$C_{\mathrm{latency}} \;=\; (L_{\mathrm{max}} - L_{\mathrm{maj}}) \cdot \ell,$$

where $\ell \in \{\ell_{\mathrm{EP}}, \ell_{\mathrm{TP}}\}$ is the per-token decoding latency.

**Batch Size Assumption.** Unless otherwise specified, we assume $N = 8192$, consistent with large-batch inference configurations reported in recent MoE serving studies.

This refined two-phase model enables a consistent estimation of total cost under static EP, static TP, and our adaptive approach, which combines both modes depending on the stage of execution.

**Estimated Results** The estimated results in Figure 6 demonstrate that our adaptive approach (ParaSMoE) consistently reduces rollout cost compared to static EP or DP×TP across a wide range of workload configurations. The improvement is particularly pronounced when the major length is much smaller than the maximum length, where ParaSMoE achieves up to $3.75\times$ speedup by exploiting EP's high throughput before the major length and TP's lower latency in the tail. Even in less favorable cases with relatively balanced lengths, ParaSMoE still provides a minimum of $1.41\times$ speedup over static baselines. These results highlight that adaptively switching parallelism is effective not only in extreme long-tail scenarios but also in moderate ones, ensuring robust efficiency gains. More generally, when both the major length and the maximum length are long, EP tends to be the most effective due to its throughput advantage; when both are short, TP is preferred due to its lower latency; and when there is a large gap between the two, ParaSMoE achieves the best trade-off by combining the strengths of both strategies.

