# OpenReview forum: "ParaSMoE: Enabling Parallelism Hot-Switch for Large Mixture-of-Experts Models"
_ICLR.cc/2026/Conference — ICLR 2026 Conference Withdrawn Submission_

### Official Review · Reviewer_eyW3 · 2025-10-20

**Soundness:** 2
**Presentation:** 3
**Contribution:** 2
**Rating:** 2
**Confidence:** 4

**Summary:**

This paper introduces EP-TP hot-switch to get the best parallelism under different workload (batch size) to get the best performance for the MoE model. The paper's key observation is that TP should be used when batches are small, while EP should be used when batches are large. This workload-specific hot-switching strategy can achieve better performance than a single parallelization strategy. Building on this insight, the paper proposes and implements a specific hot-switching method, introducing pipelined execution to mitigate the overhead of parallelization strategy transitions.

**Strengths:**

1. The motivation of paper (hot-switch to get the best parallelism under different workload) seems basically reasonable (although it has already been found in previous work).
2. The writing is good and easy to understand.

**Weaknesses:**

1. **Lack of Important Related-work:** The insight that TP can be better than EP in certain scenarios has already been proposed in MegaScale-MoE[2], but MegaScale-MoE is not cited in this paper.
2. **Critical Lack of Ablation Studies.** Figure1 shows TP vastly outperforming EP on small batches. The paper's only explanation is a single, superficial sentence (Line 139-141). This is insufficient for an ICLR publication. If the authors really think that the gap between EP and TP comes from communication, then the authors should conduct ablation studies and give a detailed analysis. For example, how much does the latency of cross-node communication differ from that of on-device latency, and what is the specific contribution of this difference to overall performance? Beyond that, I believe the performance gap between EP and TP is likely related to multiple factors, such as DP redundancy and computational disparity caused by imbalance, but these have not been discussed.
3. **Indirect and Overstated Evaluation of the Primary Use-Case:** The paper's claim of a 1.4-3.7x speedup for RL rollout (Line 91) is derived entirely from a simplified estimation. This is a significant weakness, as it overstates the work's empirical contribution. Without integration and validation in a real RL framework, the claimed speedup remains a projection, not a measured result.
4. **The evaluation is very weak:** The main evaluation only includes a weight re-sharding overhead analysis (Figure 4). I didn't see any online/offline serving experiments report metrics such as TTFT/TPOT. The setting of the motivation experiment is also strange: Figure 1 includes the setting of EP16 and TP8, TP8xDP2, where are EP8 and EP8xDP2?

**Questions:**

1. Could you explain the performance gap in the motivation example in detail? For example, why is EP16 worse than TP8? I'd like to see a comprehensive profile rather than a simple qualitative guess.
2. How is the EP implemented in the paper? Are optimization approaches such as DeepEP[1] applied?
[1] https://github.com/deepseek-ai/DeepEP
[2] https://arxiv.org/abs/2505.11432

---

### Official Review · Reviewer_cp3N · 2025-11-01

**Soundness:** 3
**Presentation:** 3
**Contribution:** 3
**Rating:** 2
**Confidence:** 4

**Summary:**

The paper introduces ParaSMoE that uses an efficient hot-switch mechanism to seamlessly transitions between Expert Parallelism (EP) and Tensor Parallelism (TP), to adapt the parallelism strategy to workloads. This is motivated based on the following use-cases: (1) Some requests with strict low-latency requirement cannot be satisfied by an EP serving instance, while deploying too many TP units can be wasteful under high request concurrency in general situations; (2) For batch generation for rollout in reinforcement learning, as generation progresses sequences complete at different lengths, causing the effective batch size to gradually shrink. Thus, a batch generation could start in the high-throughput EP configuration, and dynamically switch to the configuration of multiple lower-latency TP instances.

**Strengths:**

- The idea is well-motivated and can "potentially" significantly speed up inference and RL. I say "potentially", as the paper lacks experimental evaluations.

- The paper is self-contained and easy to follow and is generally well-written. I enjoyed learning about some lower-level details.

**Weaknesses:**

- A paper that aims to improve inference "efficiency" must contain extensive experiments to show the gains in various use-cases. However, the paper lacks any kind of such experiments. The only evaluation is provided in Fig 4, but it simply shows the latency of hot switching. No efficiency gains are provided for the use-cases motivated in the paper (that I mentioned in my summary). While I was excited about the idea, I got very disappointed by the lack of empirical evaluations in the experiments section. For this idea to have an impact, the authors should show its applicability across (1) a few different LLMs, (2) in different scenarios in particular RL (for an ML conference). The paper mentions "estimated" RL efficiency in the appendix, which is based on numbers reported by SGLang. I didn't find this convincing as usually there is a big gap between such estimations and what happens in practice (to my own experience). In particular, such estimations should not be mentioned in the abstract with such a strong language, as it's not confirmed by experiments.

- Without the RL experiments, why do the authors think this paper is more suited for an ML conference rather than e.g. MLSys? I'm not against publication in an ML conference, but I'd really expect to see more ML connections for the idea to be influential and well-received by the ML community.

**Questions:**

Can you provide real inference speedup evaluations for (1) a few different LLMs, and (2) show the benefit to RL training? If so, I'd be happy to raise my score.

---

### Official Review · Reviewer_QE2U · 2025-11-06

**Soundness:** 2
**Presentation:** 2
**Contribution:** 2
**Rating:** 4
**Confidence:** 2

**Summary:**

This work proposes a scheme for switching between EP and TP.

**Strengths:**

1. The "Hot-Switch" mechanism allows a model to switch from EP to TP (or vice versa).
2. The paper is written very clearly.
3. The research topic is important, although it focus on computer systems and high performance computing than machine learning algorithms.

**Weaknesses:**

1. This work was only tested on two hardware devices(A100, H200).
2. No more models were tested.
3. The switching timing is manually set.
4. There was no test under Heterogeneous GPU Cluster, which is quite common.

**Questions:**

1. Once a switching error occurs (data misalignment or communication asynchrony), troubleshooting becomes extremely complex. How can the paper improve its adaptability?

2. Is it possible to avoid manually setting the switching conditions when switching?

---

### Official Review · Reviewer_BXjf · 2025-11-09

**Soundness:** 2
**Presentation:** 2
**Contribution:** 2
**Rating:** 2
**Confidence:** 4

**Summary:**

The paper proposes ParaSMoE, the first work to enable efficient switching between Expert Parallelism (EP) and Tensor Parallelism (TP) for Mixture-of-Experts (MoE) inference, allowing for seamless adaptation to the optimal configuration based on varying workloads. This dynamic capability addresses applications like MoE serving with differing latency requirements and the decrease in batch sizes observed during Reinforcement Learning (RL) rollouts. This scheme minimizes stall time by pipelining the inter-node All-Gather (over slower RDMA), intra-node All-to-All (over fast NVLink), and weight permutation (on SMs), effectively hiding the long latency of inter-node transfers. The experiements demonstrate the capability of converting Qwen3-235B MoE model within 0.7 seconds, and achieve 1.4-3.7x estimated speed up in RL rollout.

**Strengths:**

The authors claim to be the first to enable practical, efficient, dynamic hot-switching between EP and TP configurations. This capability is highly relevant to common real-world issues, such as adapting MoE serving to varied Quality-of-Service (QoS) latency requirements and the dynamic batch-size changes in RL rollouts. The implementation methodology are adequately presented and sufficient to grasp the overall idea.

**Weaknesses:**

See questions below.

**Questions:**

1. Aside from the hot-switching latency experiments (Figure 4), the performance benefits claimed for real-world applications (MoE serving QoS and 1.4-3.7x speedup in RL rollout) are based on a cost model estimation. Could the authors provide experimental benchmarks of ParaSMoE's end-to-end performance on a full RL rollout workload or a dynamic MoE serving environment on real hardware? In addition, could the authors provide the source code for verification and reproducibility?

2. Figure 1's split presentation of the EP/TP trade-off is confusing. Could the authors condense this data into a single Latency vs. Throughput plot (potentially with log-scale) to clearly illustrate the Pareto Frontier, which is conventional for analyzing parallelism efficiency? Furthermore, could the authors ensure all data points on the plot are clearly labeled with their specific configuration (EP/TP/DP values and batch size)?

3. The descriptions of hardware constraints (e.g., NVLink limited to 8 GPUs) and the EP/TP trade-off (Lines 75-77) are potentially confusing. Could the authors revise these sections to explicitly confine hardware claims to the DGX A100/H200 architectures used in the experiments? Regarding the trade-off, could the language more clearly distinguish between the small-batch regime (where TP excels) and the high-concurrency regime (where EP excels)? The current description is:

   > EP naturally scales to multiple GPUs, enabling it to sustain much higher batch size and provide better throughput under heavy request concurrency. However, this design incurs cross-server communication overhead, leading to higher per-token latency at small batch sizes. In contrast, TP excels in small-batch scenarios where it offers both lower latency and higher throughput within a single server. But its scalability is fundamentally constrained by limited device memory and significant communication costs under high request concurrency, making TP less efficient than EP.

   For system analysis, performance optimization typically targets the Pareto Frontier of latency versus throughput . In the context of MoE, a configuration heavily optimized for Tensor Parallelism (TP) generally sits toward the low-latency end of this frontier, but offers lower scalable throughput. Could the authors revise the text to explicitly clarify how ParaSMoE's dynamic switch between EP and TP allows the inference system to continuously operate along this Pareto Frontier as the workload profile (specifically, the batch size) changes? Furthermore, optimizing resource utilization requires the configurations (EP, TP, DP, and Batch Size) to adapt accordingly, and the current plot in Figure 1 does not fully illustrate this necessary adaptation of configurations.

4. The text states that intra-node communication via NVLink "cannot expand to more than 8 GPUs". This claim is generally inaccurate, as modern architectures (e.g., NVIDIA GB200 NVL72) utilize NVLink Switch technology to achieve cross-node high-speed networking across 72 GPUs, with potential scaling up to 576 GPUs with fast P2P connections.

   Could the authors confine this specific constraint on NVLink expansion to the experimental hardware (e.g., DGX A100 or DGX H100 nodes) used to maintain technical accuracy and avoid over-generalization? The original description reads:

   > Intra-node communication is through high-speed networsk like NVIDIA NVLink, but cannot expand to more than 8 GPUs. Inter-node communication (via Infiniband or Amazon EFA) is significantly slower than intra-node, but can expand to a large number of nodes and GPUs.

Minor issues:
- Line 101: "high-speed networsk" -> "high-speed network"
- Line 130: "following a data parallel pattern (with optional )" ?

---

### Official Review · Reviewer_AoGE · 2025-11-12

**Soundness:** 2
**Presentation:** 2
**Contribution:** 2
**Rating:** 4
**Confidence:** 3

**Summary:**

This paper introduces ParaSMoE, a system designed to enable hot-switching between Expert Parallelism (EP) and Tensor Parallelism (TP) in large-scale Mixture-of-Experts (MoE) models. The proposed approach reshards model parameters dynamically at runtime, allowing the system to transition between parallelization strategies in approximately 0.7 seconds, without the need for model reloading or process restart.
The authors demonstrate that this mechanism can enhance inference efficiency in two representative scenarios:
(1) MoE serving, where latency and throughput requirements fluctuate with varying request loads, and
(2) reinforcement learning rollouts, where batch sizes typically shrink over time.
From a systems perspective, ParaSMoE achieves this efficiency by pipelining inter-node All-Gather operations (over RDMA) with intra-node All-to-All transfers (via NVLink) and on-device weight permutations, thereby mitigating the latency associated with cross-node communication.

**Strengths:**

ParaSMoE is a practical system that enables runtime switching between TP and EP for large-scale MoE inference. This capability is highly relevant for modern inference workloads where latency and batch-size characteristics vary dynamically.
The paper aptly identifies practical challenges encountered in real deployments—namely, the latency constraints arising from small-batch inference during serving, and the progressive reduction of batch sizes in reinforcement learning rollouts. Both cases are well-motivated examples where dynamic switching between parallelization strategies can provide tangible benefits.

**Weaknesses:**

The paper presents a technically interesting idea, but lacks sufficient end-to-end evaluation.
While the switching latency (≈0.7 s) is convincing, the reported performance gains in RL rollouts are based only on analytical models, without real benchmarks for serving or RL workloads.
The experiments are limited to Qwen3-MoE models on 8–16 GPU nodes, leaving scalability to larger or heterogeneous clusters untested.

The paper also does not specify when or how the switch between EP and TP should occur; no automatic policy or workload-driven mechanism is provided.
Some hardware assumptions require clarification—e.g., the NVLink limit of eight GPUs applies only to the DGX A100/H200 setup—and the reliance on ultra-fast RDMA may limit generality.
Robustness aspects such as fault recovery during switching are not discussed.

Finally, Figure 1 could more clearly illustrate the latency–throughput trade-off in a single Pareto plot, and a few minor textual issues remain.

**Questions:**

1. Could you provide or plan to provide empirical results of ParaSMoE on a complete RL rollout or dynamic MoE serving workload? Specifically, how does hot-switching impact total task completion time and throughput in a real distributed setup?
2. How is the switching decision (EP ↔ TP) made in practice? Is there a heuristic, threshold, or workload profiler used to trigger the switch? If not, how would you envision integrating an automatic policy into ParaSMoE?
3. Have you tested ParaSMoE on larger or heterogeneous clusters (e.g., more than 16 GPUs, or mixed A100/H100 systems)? If not, do you foresee any bottlenecks or synchronization challenges that would arise at larger scales?
4. What happens if a failure occurs during the hot-switching process (e.g., interrupted All-Gather, RDMA timeout)?
Is there any mechanism to ensure consistency or rollback in such cases?

---

### Note · Authors · 2025-11-14

I have read and agree with the venue's withdrawal policy on behalf of myself and my co-authors.